

# Personalized movie recommendations based on deep representation learning

Luyao Li[1], Hong Huang[1], Qianqian Li[2] and Junfeng Man[1,3,4]

[1] Department of Computer Science, Hunan University of Technology, Zhuzhou, China
[2] Hunan University of Technology, Zhuzhou, China
[3] Department of Computer Science, Hunan First Normal University, Changsha, China
[4] Hunan Provincial Key Laboratory of Information Technology for Basic Education, Hunan First Normal University, Changsha, China

## ABSTRACT

Personalized recommendation is a technical means to help users quickly and efficiently obtain interesting content from massive information. However, the traditional recommendation algorithm is difficult to solve the problem of sparse data and cold-start and does not make reasonable use of the user-item rating matrix. In this article, a personalized recommendation method based on deep belief network (DBN) and softmax regression is proposed to address the issues with traditional recommendation algorithms. In this method, the DBN is used to learn the deep representation of users and items, and the user-item rating matrix is maximized. Then softmax regression is used to learn multiple categories in the feature space to predict the probability of interaction between users and items. Finally, the method is applied to the area of movie recommendation. The key to this method is the negative sampling mechanism, which greatly improves the effectiveness of the recommendations, as a result, creates an accurate list of recommendations. This method was verified and evaluated on Douban and several movielens datasets of different sizes. The experimental results demonstrate that the recommended performance of this model, which has high accuracy and generalization ability, is much better than typical baseline models such as singular value decomposition (SVD), and the mean absolute error (MAE) value is 98%, which is lower than the best baseline model.

Corresponding author
Junfeng Man, mjfok@qq.com

## INTRODUCTION

The explosion of information on the Internet has become one of the major obstacles to extracting useful information efficiently from available databases (*Cui et al., 2020*). How to get the correct information quickly and efficiently is a key issue in the development of the Internet. However, users may be able to access the information they need from an information-rich website thanks to recommendation algorithms. The recommendation system's primary component is the recommendation algorithm. The current recommendation algorithm is mainly divided into the following two types. The first is the collaborative filtering algorithm, including the memory-based collaborative

filtering algorithm and the model-based collaborative filtering algorithm. The second is the recommendation algorithm based on deep learning. In the context of sparse data, collaborative filtering-based recommendation algorithms cannot efficiently extract deep structural features of users and items. Deep learning-based recommendation systems are getting more attention because they can make recommendations in certain real-world situations more quickly and accurately (*Qin & Zhang, 2021*; *Paul et al., 2016*).

Model-based collaborative filtering is widely used among traditional collaborative filtering algorithms. Most of them use machine learning algorithms to carry out data mining, such as the clustering model (*Feng, Zhao & Zhou, 2020*), matrix factorization model (MF) (*Mandal & Maiti, 2020*), regression model (*Kouadria, Nouali & Al-Shamri, 2020*), Bayesian model (*Sun et al., 2021*), *etc*. Scholars in this domain have performed comprehensive study, which has yielded excellent results. For example, (*Wang et al., 2021a*; *Wang et al., 2021b*) proposed a collaborative filtering algorithm based on trust, which added user-item trust records to rating information to reduce the impact of data sparsity. This model increases the robustness and accuracy of the collaborative filtering algorithm. *Tao, Gan & Wen (2019)* proposed a collaborative filtering algorithm combining item similarity calculation based on the ALS-based collaborative filtering algorithm. When the number of users is greater than the number of items, the similarity calculation of items can reduce the loss of hidden information, indirectly solve the problem caused by sparse data, and improve the scalability and accuracy of the recommendation algorithm. *Thakkar et al. (2019)* proposed a combination of two collaborative filtering algorithms, combining multiple linear regression (MLR) and support vector regression models for item-based collaborative filtering and user-based collaborative filtering. The algorithm sends the user-item matrix to the item-based collaborative filtering and user-based collaborative filtering algorithms for prediction and feeds the predictions as training data to the MLR and support vector regression models for final prediction. The research shows that this model can reduce prediction errors. *Feng et al. (2020)* proposed the fusion collaborative filtering model (FPMF) to solve the problem of data sparsity, that is, the user's local neighborhood information is fused into the global rating information through matrix decomposition, which can capture the linear or nonlinear relationship generated by the user's extreme behavior. Constrained by similarity, the model can effectively mine the potential behavior of users, and improve the accuracy of the algorithm. The above algorithms based on collaborative filtering have a wide range of applications. However, the collaborative filtering-based recommendation algorithm is a direct prediction using a user-item rating matrix with very low data density, so the recommendation accuracy is low in scenarios where the data set is very sparse and the user's historical behavior is very small. In addition, the collaborative filtering recommendation algorithm only uses information about the direct interaction between the user and the item and does not extract sufficient features about the user and the item. And the recommendation performance of the above algorithm obviously decreases when the data volume is large and the structure is complex.

In view of the limitations of traditional recommendation algorithms, the superior performance of deep learning in feature extraction has attracted the attention of scholars. In 2006, Hinton first proposed the concept of deep learning, which has a powerful

feature learning ability (*Wang et al., 2021a*; *Wang et al., 2021b*). Deep learning models are widely used in data feature extraction to deal with the problem of large-capacity and high-dimensional sparse data. It extracts hierarchical features from the input training set through nonlinear transformation. For example, *Liu (2018)* has proposed a collaborative filtering algorithm based on a restricted Boltzmann machine (RBM) (*Fachechi et al., 2022*), which extracts item labels in the prediction process. Users use the scored items to describe the degree of users' interest in the items, and then predict the user's liking for the unscored item, which solves the cold-start problem in traditional collaborative filtering to a certain extent and improves the accuracy of recommendation. *Xue et al. (2019)* proposed a collaborative filtering algorithm based on deep items to make recommendations. Item-based collaborative filtering uses items rated by users to model user profiles, and then makes rating predictions based on the similarity between items and profiles. However, the simple collaborative filtering algorithm only models the second-order interaction between two items. Considering the nonlinear relationship between items, we use a neural network to model the interaction between items. The shallow neural network in the above model captures the complex process of user decision-making and improves the performance of collaborative filtering. Compared with shallow neural networks, deep neural networks have the advantage of extracting complex features. With the increase of data volume and complexity, shallow neural networks are difficult to extract hidden features. To overcome the challenge of the shallow neural network on data modeling, relevant scholars proposed to use of deep neural architecture to extract higher-level features, so as to improve the modeling ability of the model. Deep-structured neural networks are an algorithm of deep learning. In recent years, due to its powerful adaptive and nonlinear feature extraction capabilities, deep structure models have been proposed to extract hidden features and construct representations on large-scale data sets (*Ahmadian, Ahmadian & Jalili, 2022*). For example, *He et al. (2017)* proposed neural network-based collaborative filtering (NCF), which uses multi-layer perceptron to model the interaction function between users and items and extract the nonlinear relationship between user and item. Experiments have proved that deep-level neural networks have better recommendation performance. *Covington, Adams & Sargin (2016)* also used a three-layer perceptron model to model the feature representation of the user and the item, and then used softmax regression to derive the probability of the user viewing the video. However, the cost of using multilayer perceptrons to make recommendations in real industrial environments is higher (*Rendle et al., 2020*). *Rassweiler Filho, Wehrmann & Barros (2017)* proposed a new approach to DeepRecVis that extracts deeper features from user-item data, applying the features proposed by the new model to a content-based recommendation algorithm that extracts enough semantic representations from items to make effective recommendations. But content-based recommendations ignore hidden connections between users and are more susceptible to the cold-start factor. *Bi, Liu & Fan (2020)* proposed to build a regression model for user rating prediction based on a deep neural network, which uses basic user data and item data to improve recommendation performance. But due to the growing concern about privacy, such attempts have been rare.

To solve the above problem, we propose a model combining DBN (*Naskath, Sivakamasundari & Begum, 2022*) and negative sampling softmax for a recommendation, and apply it to the movie recommendation field. First, we process the dataset into a user-item rating matrix and then feed the user-item rating matrix into the model for deep feature extraction. In the real world, the user-item matrix is usually sparse. The deep neural network architecture allows for sufficient semantic learning of the data to effectively extract hidden features and build a more robust representation of user-item features. In addition, compared to the sigmoid function in the DBN model, this article uses the softmax function for multiclassification to calculate the probability of a user watching a movie. The efficiency of the recommendation system is also crucial, and finally negative sampling softmax is used to improve the computational efficiency of the model.

The work of this article is as follows:

(1) In this article, a DBN based on an RBM was used to extract hierarchical features from the movie user-item rating matrix. In the construction of a multi-layer RBM, representation learning was used for dimensionality reduction processing to transform the sparse matrix into a dense matrix, which alleviated the data sparsity and cold-start problem in traditional collaborative filtering.

(2) Since the sigmoid function in the DBN is more suitable for binary classification problems and may output multiple parallel results, softmax is adopted as the last layer output in this model, and the prediction problem is regarded as a multi-classification problem. The softmax layer categorizes the DBN's user and item feature statistics and calculates the probability of each user seeing each movie. In addition, the negative sampling mechanism is used to improve the computational efficiency of the model. Compared with sigmoid as the output, this method can improve efficiency better and reduce the MAE value by 91%.

(3) In this article, two experiments demonstrate the effectiveness of the proposed algorithm. Firstly, the comparison experiment with the other eight models on the MovieLens 1M dataset shows that the recommendation performance of the proposed model is better than the traditional recommendation algorithm. Secondly, the experiment on the MovieLens dataset with different data sizes verifies the stability of the proposed model. The experimental results indicate that the proposed model trained on the Douban dataset has also high accuracy, which indicates that the method adopted in this article has a strong generalization ability.

The rest of the article is organized as follows: 'Theoretical Background' describes the relevant theoretical background. The 'Recommendation Algorithm' section introduces the recommendation algorithm and the function of each component. In 'Experimental Analysis', the proposed model and the comparison model are analyzed experimentally on the data sets Movielens and Douban. In 'Conclusion', a brief summary of this article and the direction of future research are pointed out.

**Table 1  Main symbols' definition.**

| Symbol | Definition | Symbol | Definition |
|---|---|---|---|
| $a_i$ | Visible units bias | $Z$ | Training set |
| $b_j$ | Hidden units bias | $x$ | Training data |
| $w_{ij}$ | Weight between visible units and hidden unit | $y_i$ | Label of training data |
| $V_i$ | Text feature state of visible unit i | $k$ | The rating of movies |
| $h_j$ | Text feature state of hidden unit j | $\theta$ | Function parameter |
| $S$ | Activation probability | $M$ | Number of movies |
| $\Delta w_{ij}$ | Updating the value of the weight | $N$ | Number of users |
| $\Delta a_i$ | Updating the value of visible units | $p(y = j\|x)$ | Probability of each x belonging to each category j |
| $\Delta b_j$ | Updating the value of hidden units | $j$ | Number of categories |
| $\varepsilon$ | Learning rate | | |
| $<.>_{data}$ | Value of visible units i multiplied by hidden units j before reconstruction | | |
| $<.>_{recon}$ | Value of visible units i multiplied by hidden units j after reconstruction | | |

# THEORETICAL BACKGROUND

## Restricted Boltzmann machine

The RBM is composed of a visible layer and a hidden layer, with a connection within the layer and no connection between layers. As shown in Fig. 1, each layer is composed of several neurons, and each neuron is a binary variable with a value of 0 or 1. The training set is entered into the visible cell, multiplied by an independent weight when passed to the hidden cell, plus a bias item, and finally entered into the sigmoid activation function. The calculation from the visible layer to the hidden layer can be regarded as the coding process of RBM, while the reconstruction process is the decoding process of the hidden layer. In the decoding process, the hidden layer can be regarded as the visible layer, and the output of the activation function can be regarded as the input of the visible layer. In the output of the decoding layer, we get the reconstructed value $r$. By iterating the training parameters w, a, and b continuously, the error between the reconstructed values and the input is minimized. The reconstructed value of the network is the prediction score, and the maximum predicted value can be selected to represent the probability of user and item interaction. The fast learning algorithm based on contrast scatter (CD) is adopted to improve the training efficiency of the model (*Fachechi et al., 2022*). The main symbol definitions in this article are shown in Table 1.

Figure 1 shows the structure of a two-tier RBM. The term a is the visible layer's bias term, w is a weight matrix, and b is the hidden layer's bias term. For a given set of states (v,h), the energy function of RBM is expressed as the formula (*Hinton, 2012*) (1).

$$E(v, h) = -\sum_{i=1}^{n} a_i v_i - \sum_{j=1}^{m} b_j h_j - \sum_{i=1}^{n}\sum_{j=1}^{m} v_i w_{ij} h_j \tag{1}$$

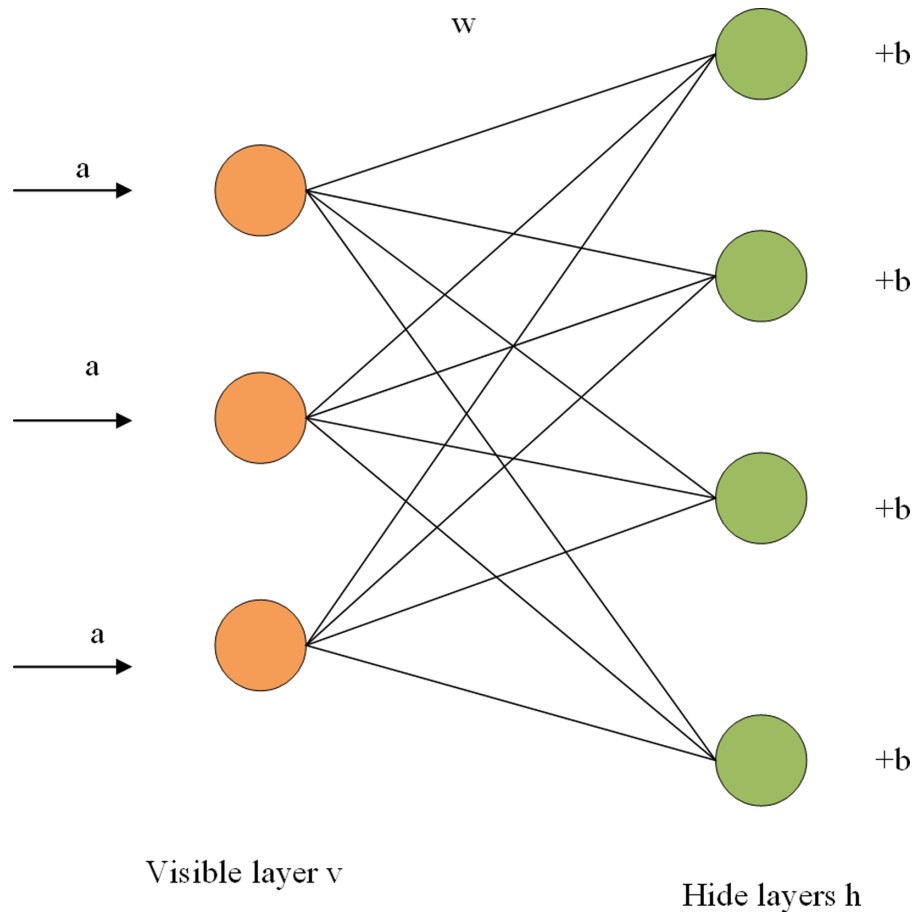

Visible layer v

Hide layers h

**Figure 1  RBM structure diagram.**

Where $a_i$ represents the bias of the visible unit, $v_i$ is the $i$th visible unit, $b_j$ represents the bias of the hidden layer, $h_j$ represents the $J$th hidden unit, and $w_{ij}$ represents the weight between the $i$th visible unit and the $j$th hidden unit. The activation probability of visible and hidden elements can be obtained from the energy function of RBM, as shown in formulas Eqs. (2) and (3) (*Zhang et al., 2022*).

$$P(h_i = 1|v) = sigmoid(w_{ij}v + b_i) \tag{2}$$
$$P(v_j = 1|h) = sigmoid(w_{ij}h + a_i). \tag{3}$$

Here, the sigmoid activation function is adopted, which can be obtained from formula (4) (*Dubey, Singh & Chaudhuri, 2022*).

$$S = \frac{1}{1 + e^{-x}} \tag{4}$$

The parameters of the energy function are $w_{ij}$, $a_i$, and $b_j$, and the weight is updated and learned by the fast learning algorithm of contrast divergence (CD), such as in formulas Eqs. (5), (6) and (7) (*Song & Chen, 2022*). The main process of the fast learning algorithm with contrast divergence is shown in Algorithm 1.

$$\mathbf{V}_{w_{ij}} = \varepsilon(<v_i h_j>_{data} - <v_i h_j>_{recon}) \tag{5}$$

$$\mathbf{V}_{a_i} = \varepsilon(<v_i>_{data} - <v_i>_{recon}) \tag{6}$$

$$\mathbf{V}_{b_j} = \varepsilon(<h_j>_{data} - <h_j>_{recon}) \tag{7}$$

---

**Algorithm 1:** CD algorithm

---

**Input:** Z=$\{x_1, x_2, ... x_n\}$ training data, $v_i$ is the visible layer unit, $h_j$ is the

hidden layer unit. The learning rate is $\epsilon$

**Output:** Model parameters $\theta = \{w, a, b\}$

1  Initialize model parameters randomly $\theta$

2  **For** all hidden units i **do**

3      calculate $p(h_{1i} = 1|v_1)$ according to Eq.(2)

4  **End for**

5  **For** all visible units j **do**

6      calculate $p(v_{2j}|h_1)$ according to Eq.(3)

7  **End for**

8  **For** all hidden units i **do**

9      calculate $p(h_{2i} = 1|v_2)$

10  **End for**

11  $w \leftarrow w + \varepsilon(h_1 x_1' - p(h_2 = 1|v_2)v_2')$

12  $a \leftarrow a + \varepsilon(v_1 - v_2)$

13  $b \leftarrow b + \varepsilon[p(h_1 = 1|v_1) - p(h_2 = 1|v_2)]$

---

## Deep belief networks

A DBN is composed of a multi-layer RBM (*Tang & Zhang, 2022*) which is a typical deep learning architecture. The experimental results of DBN and RBM on the same data set show that the modeling ability of deep network structure is better than that of shallow network structure (*Le Roux & Bengio, 2008*). It is shown that unless RBM has perfectly modeled the data, adding hidden units will result in strictly improved modeling capability. The DBN can obtain more abstract and advanced feature representation through the representation learning of multiple hidden layers. In addition, the feature extraction ability of sparse data is also very strong. So this article uses a DBN to extract the feature of data. The structure of the DBN is composed of several small modules. The simple network structure diagram is shown in Fig. 2. The internal structure of DBN is trained using a bottom-up greedy

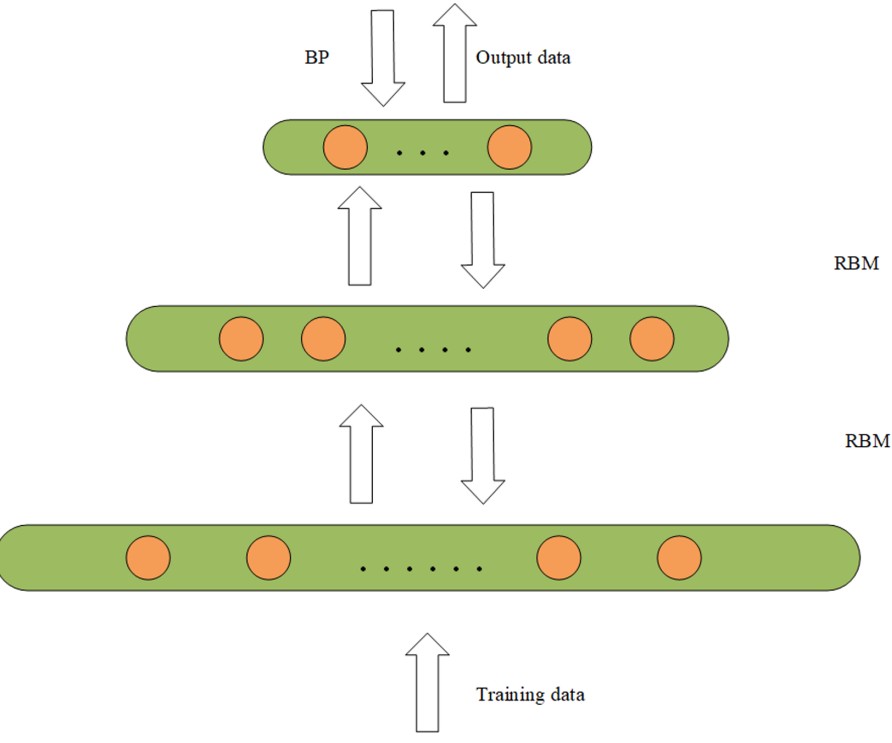

**Figure 2  DBM structure diagram.**

learning algorithm. The data is fed into the first layer of the RBM for training to obtain w and b for the first layer, and then the output data representation is fed to the second layer of the RBM for training, repeating the above steps until one is satisfied with the number of layers. Where the hidden layer acts as the visible layer of the next RBM module, and the output of the activation function acts as the input of the visible layer. The final stage is to fine-tune the model by backpropagation to improve its performance. The training of DBN is independent training from bottom to top (*Tian et al., 2022*), that is, the locally optimal solution is first achieved, and then fine-tuned by backpropagation.

## Representation learning

Representational learning is a feature representation method of input data, which can be classified according to supervised and unsupervised learning. This article adopts unsupervised representational learning. Representation learning obtains more effective data representation by learning the input training data set. In the field of recommendation algorithms, it is very important to obtain a more representative feature representation for solving sparse and complex data problems. With the development of technology, deep learning has great potential to obtain features from data, so deep learning technology has been widely used in recommended fields (*Ahmadian, Ahmadian & Jalili, 2022*). RBM is a type of unsupervised learning in representation learning. It is composed of two layers of undirected network structure, and its hidden layer is mainly used to obtain higher-level feature representation of data. The DBN is trained using an unsupervised learning method,

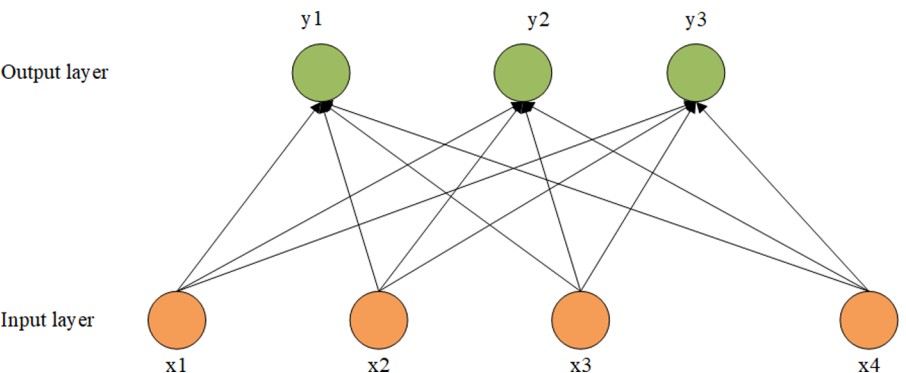

**Figure 3  Structure diagram of softmax.**

and the locally optimal solution is obtained layer by layer from bottom to top. Finally, it is fine-tuned and optimized by backpropagation. Since the adopted data set may have problems such as high dimension and sparse data, dense and low-dimensional feature representation can be obtained through representation learning of multiple hidden layers, and the deep network structure can mine potential feature representation of data through representation learning. This is to solve the data sparsity problem of traditional collaborative filtering.

### Importance sampling softmax

Softmax regression is suitable for multi-classification problems in natural language processing, and the output is a probability distribution. Softmax regression is equivalent to a single-layer neural network, assuming that our training set is $Z = \{(x_1, \ldots, x_n)\}$, where $x_i$ ($i \in \{1,2,\ldots,n\}$). The training set is labeled $y_i$ ($i \in \{1,2, \ldots,m\}$), assuming that there are $K$ categories in the training set, namely $y_i \in \{1,2, \ldots,k\}$. The softmax layer calculates the probability that the training set belongs to each category. As shown in Fig. 3, we input our training set x into the input layer and get the probability of each category through the softmax function. Formula (8) is to calculate softmax regression probability (*Yao & Wang, 2019*). Where $\theta$ is a parameter of the softmax regression model, $e^{\theta_j^T x_i}$ prevents negative values of probability, and $\frac{1}{\sum_{j=1}^{k} e^{\theta_j^T x_i}}$ makes the probability values of all output layers add up to one.

$$p(y_i = j | x_i; \theta) = \frac{e^{\theta_j^T x_i}}{\sum_{l=1}^{k} e^{\theta_j^T x_i}} \tag{8}$$

## RECOMMENDED ALGORITHM

In this article, the unlabeled data set is trained by a DBN. The internal structure is optimized by the bottom-up greedy learning algorithm. The last layer is fine-tuned and optimized by

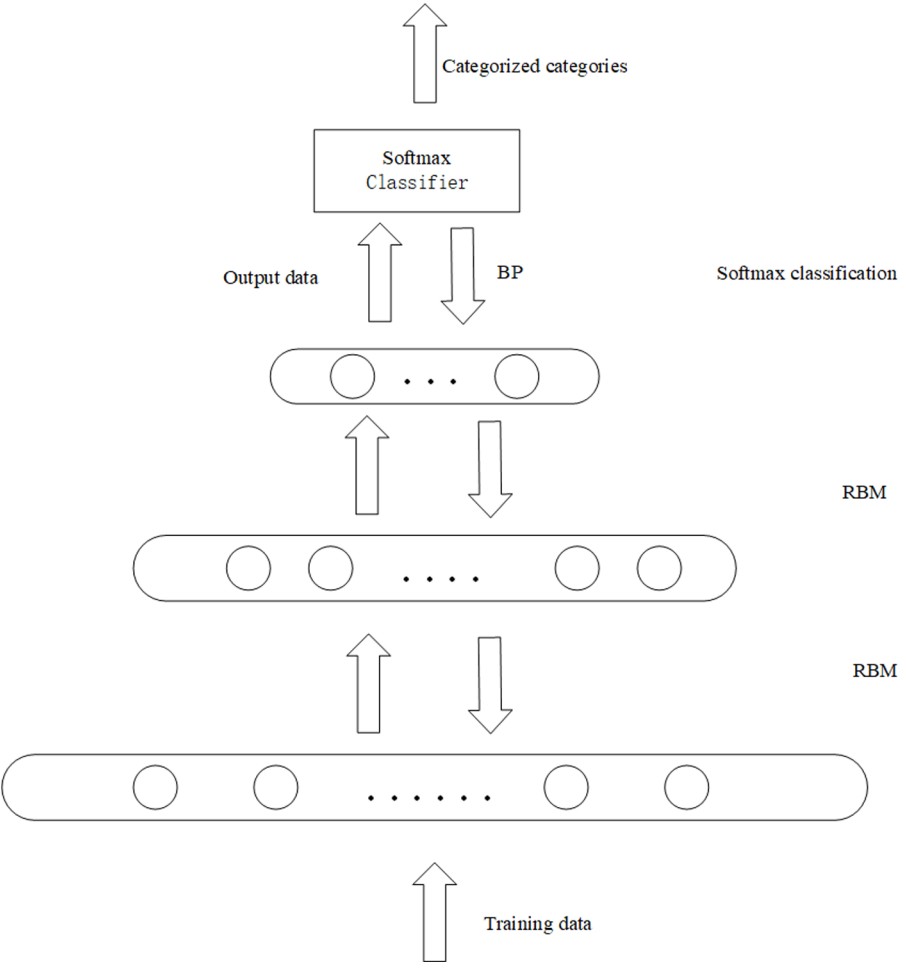

**Figure 4** **The framework of the proposed recommendation algorithm.**

backpropagation. Finally, the probability is predicted by softmax. The model structure is shown in Fig. 4.

The model uses an unlabeled dataset to train the DBN model. The input layer of the DBN, the visible layer, is composed of M softmax units. M is the number of movies in the user-item rating matrix, and N is the number of users with a movie score of 1 to k. Each user in the input layer can be viewed as an RBM instance, and there are as many softmax visible cells as there are movies rated by the active user. Since different users rate different movies, each user has different visibility units, and softmax does not have visibility units for movies without user ratings. When different users rate the same movie, their softmax has the same weight between the visible unit and the hidden layer (*Nazari, Koohi & Mousavi, 2022*).

The rating matrix of users and movies is a high-dimensional sparse matrix, so to overcome the challenge of extracting data features from a matrix with sparse data, this article uses a deep network structure for representation learning to extract data features. We

change the sparse matrix into a dense matrix by reducing dimension. In the first layer, we use the hidden layer with 1,500 hidden units and reduce the original input data dimension to 1,500. At this time, our model is a simple shallow structure RBM. As mentioned in the first section, the modeling ability of the deep network structure is much higher than that of the shallow network structure, so the hidden layer with 700 hidden units is adopted in the second layer to mine the essential features and potential information of the original dataset.

Since there is no behavior data of this user in the user-item rating matrix when a new user comes in, his user rating vector is 0. The model performs representation learning on the input user-item rating matrix. Firstly, the sparse user-item rating matrix is transformed into a dense user-item rating matrix by the way of dimension reduction. Secondly, feature learning is carried out on the dense user-item rating matrix. Finally, through the softmax layer classify, and then get the user's movie recommendation table. At this time, because the data of the new user is 0, the new user is recommended according to the first several popular recommended movies in the user's movie recommendation table, which solves the problem of cold-start of the new user.

Given the representation of user and item feature statistics, we now treat the prediction problem as a multi-classification problem. In a DBN, the sigmoid function is used for classification. The output results of sigmoid classification are independent of each other, and there may be multiple parallel results (*Wang et al., 2021a*; *Wang et al., 2021b*). Softmax function for multi-category problems, where each item is set as a category and the goal is to get the probability of the user on each category. The model wants to identify each type of movie, so softmax is used to predict the rating of the movie and thus get the probability of each user watching the movie. For softmax regression, assume that the input training data is $Z=\{(x_1, \ldots, x_n)\}$, $x_i$ ($i \in \{1,2,\ldots,n\}$). The softmax function is to calculate $P(y=j|x)$, namely for input $x$, $y$ belongs to the category of probability $j$. There are $k$ categories of $y$, namely $y_i \in \{1,2, \ldots,k\}$. The output of softmax regression is shown in Formula (9) (*Ng et al., 2012*).

In the formula, $\theta$ is the parameter of softmax regression, $\frac{1}{\sum_{j=1}^{k} e^{\theta_j^T x_i}}$ which is to achieve the probability sum of the softmax regression output to be 1, and the probability of each user watching each movie is at [0,1]. The softmax regression cost function is shown in Formula (10) (*Ng et al., 2012*).

$$p(y_i = j|x_i;\theta) = \frac{e^{\theta_j^T x_i}}{\sum_{l=1}^{k} e^{\theta_j^T x_i}} h_\theta(x_i) = \begin{bmatrix} p(y_i = 1|x_i;\theta) \\ p(y_i = 2|x_i;\theta) \\ p(y_i = 3|x_i;\theta) \\ \mathbf{M} \\ p(y_i = k|x_i;\theta) \end{bmatrix} = \frac{1}{\sum_{j=1}^{k} e^{\theta_j^T x_i}} \begin{bmatrix} e^{\theta 1^T x_i} \\ e^{\theta_2^T x_i} \\ \mathbf{M} \\ e^{\theta k^T x_i} \end{bmatrix} \quad (9)$$

$$L(\theta) = -\frac{1}{n} \left[ \sum_{i=1}^{n} \sum_{j=1}^{k} 1\{y_i = j\} \log \frac{e^{\theta_j^T x_i}}{\sum_{l=1}^{k} e^{\theta_i^T x_i}} \right] \quad (10)$$

**Table 2  Compare the efficiency of the two models under different data sets, and the unit of data is seconds.**

| Model | Movielens 100k | Movielens 1M |
|---|---|---|
| DBN+sigmoid | 82 | 916 |
| DBN+softmax | 44 | 695 |

The output of the DBN in this model is used as the input of the softmax layer, and the feature statistics of the users and items are sent to the softmax model for training. The probability of each output category can be predicted. If the prediction is consistent with the real value, the prediction is correct. In essence, it is the product of the feature vector of each user and the feature vector of the movie. The larger the product, the greater the probability of the user watching the movie. The larger the product, the greater the probability of the user watching the movie. Since there are 3,883 movies in the minimum data set we used, the denominator of softmax has to traverse all movie feature vectors, and the user's feature vector has to be dot-produced with the feature vectors of all items, which requires too much computation and may lead to low efficiency. The calculation of softmax is less computationally intensive after taking positive and negative samples. Therefore, the model uses negative sampling softmax to improve the computational efficiency, and negative sampling of the global movie can be unified with the real scene. In this article, positive and negative samples are collected based on the rating data in the dataset. Movie data below a rating of 4 are negative samples, and those greater than or equal to 4 are positive samples. To reduce error, a random selection of films with no user ratings was made. And to achieve better results, we set the number of positive and negative samples for each user to be equal. The implementation of the negative sampling softmax layer resulted in a reduction in the size of the overall prediction output and a significant reduction in time costs, as shown in Table 2. The table shows the comparison between the model with negatively sampled softmax and the model with sigmoid for two different size data sets. It can be seen that the model with negative sampling softmax takes less time to train on both datasets and is more efficient than the model with the sigmoid.

Finally, the probability of each user watching each movie is obtained through the softmax layer. In addition, the user's viewing time of the movie is used to judge whether the user has watched the recommended movie, and the viewing time of the movie that the user has not seen is set as NaN. Finally, the sorting function is used to sort according to the probability, and the recommendation probability table of the movie that the user has not watched is obtained.

The flow of the model is shown in Algorithm 2. The algorithm inputs the user-item rating matrix and outputs a list of movie recommendations. Lines 1–5 preprocess the data and set the parameters of the model. Lines 6–11 input data into the DBN model for feature extraction, and finally get the feature representation of the user and the item. Lines 13–14 represent the feature representations of users and items are sent to the softmax layer for multiple classifications, and the probability of users watching movies is obtained. After sorting, the recommendation list is obtained.

| **Algorithm 2:** DBN+softmax | |
|---|---|
| | **Input:** user rating matrix |
| | **Output:** TopN recommended movies |
| 1 | Loading the rating dataset |
| 2 | trx ← normalized users' ratings into a matrix of user-rating |
| 3 | Setting the model's parameters |
| 4 | (epochs, batches, lr) ← Initialize(setting) |
| 5 | inpx ← trx |
| | /*DBM training stage |
| | */ |
| 6 | for rbm in rbm_list do: |
| 7 | the first layer of RBM is trained using unsupervised algorithms |
| 8 | the input of the hidden layer and the output of the visible layer are calculated by the equation. |
| 9 | the visible layer is used as input to the RBM of the next layer |
| 10 | rbm.train(inpx) |
| 11 | Inpx ←rbm_output(inpx) |
| 12 | end |
| 13 | The output of DBN is used as the input of the softmax layer. |
| 14 | Output the TopN recommended movies |

# EXPERIMENTAL ANALYSIS

## Data set and implementation settings

### Data description

To prove the validity of the proposed model, we conducted experiments on the publicly available datasets Movielens and Douban. The Movielens dataset is publicly available in the mobile lens group. We used three sets of Movielens data sets with sizes of 100K,1M, and 25M respectively. The 100K dataset includes the rating data of 943 users and 1,682 movies (*Bi, Liu & Fan, 2020*), and the 1M dataset includes 6,040 users and 3,706 movies. Due to the large size of the 25M data set, part of the data was intercepted for the convenience of the experiment. The 25M Movielens dataset intercepted 10,000 users and 24,331 movies, while the Douban dataset captured 10,000 users and 1,111 movie ratings. The dataset includes the user's rating of the movie, the length of time the movie was watched, the user's ID, the movie's ID, and other fields. Each rating data is the rating of different movies by different users. The user's rating range is 1 to 5, and the larger the number is, the more the user likes the movie. Table 3 provides a detailed description of the data set.

### Experimental platform

This experiment platform uses an Nvidia GTX 1080Ti GPU. The programming tool we use is Jupiter Notebook, and the editing language is Python 3.6. Python is easy to write and highly readable, and the experimental framework is compiled based on TensorFlow 1.15. The main libraries used are Keras 2.2.4, an artificial neural network library; scikit-learn 0.24.2, which is mainly used for training test classification and calculating MAE values;

**Table 3** The statistics of the datasets.

| Dataset | Users | Items | Interactions | Density |
|---|---|---|---|---|
| Movielens100k | 943 | 1,682 | 10,000 | 0.63% |
| Movielens1M | 6,040 | 3,706 | 1,000,209 | 4.5% |
| Movielens25M | 10,000 | 24,331 | 1,496,612 | 0.62% |
| Douban25M | 10,000 | 1,111 | 17,556 | 0.16% |

pandas 1.1.5, a library for manipulating data frames; and IPython 7.16.3, a library for importing Scalable Vector Graphics (SVG).

*Parameter setting*

The setting of the hidden layer is related to the expressiveness of the model. In this experiment, the principle of halving was followed. A total of 1,500 hidden units were set in the first hiding layer of the model, and 700 hidden units were set in the second hiding layer. Finally, to obtain the feature statistics of our users and items more accurately, the data is dimensionally reduced (*Jia et al., 2022*), and the third hidden layer is set to 50 hidden units. The learning rate of the model is set to 0.075, epochs to 15, and batch sizes to 200.

*Evaluation criteria*

In this article, the mean absolute error and precision are used to evaluate the performance of the proposed model and the accuracy of the recommendation. The mean absolute error and precision are good criteria to judge the prediction. The formula for mean absolute error is as follows (11) (*Chicco, Warrens & Jurman, 2021*):

$$MAE = \frac{1}{N} \sum_{i=1}^{N} |y_i - \hat{y}_j| \tag{11}$$

Where $y_i$ is the target value, $\hat{y}_j$ isthe predicted value, and N is the number of input data. According to Formula (11), the smaller the MAE value, the better, that is, the smaller the MAE value indicates that our model is predicting better.

In addition, the validity of the model in terms of accurate prediction was assessed by calculating the precision. The formula for calculating the accuracy is shown in Formula (12), which shows the ratio of the number of movies that users are interested in to the total number of recommended movies in the recommendation list. Where TP represents the number of positive classes predicted as positive classes, and FP represents the number of negative classes predicted as positive classes.

$$\text{Precision} = \frac{TP}{TP + FP} \tag{12}$$

## Experiment analysis
### *Data preprocessing*

The loaded movie rating file dataset contains the unique ID number of the movie, the movie title, and the category to which the movie belongs; The user rating dataset stores
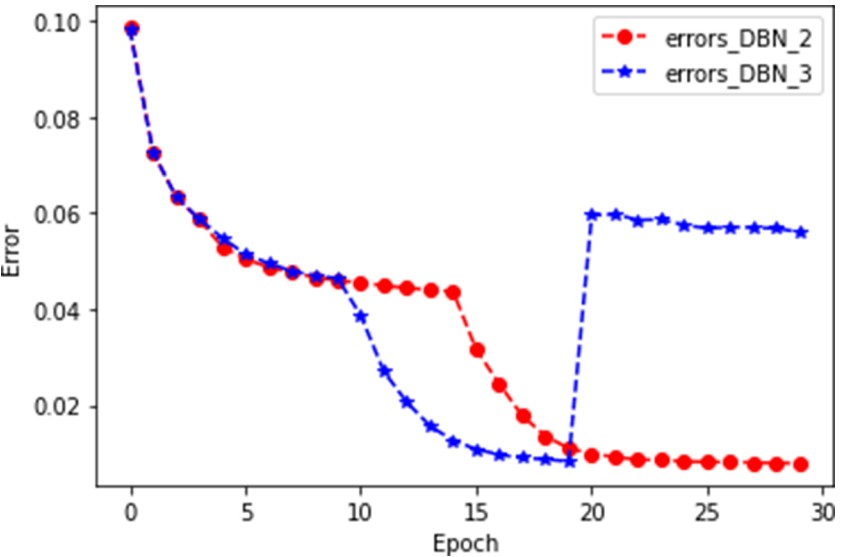

**Figure 5** Schematic diagram of the number of RBN layers.

a unique user ID number, the ID of the movie that the user has watched, and the user's rating of that movie. In order to convey the data information more intuitively, we renamed the columns in the data box and combined them with the same user ID number as the benchmark to form the final user-item rating matrix.

### Parameters of the model

In this experiment, the evaluation test was carried out on the dataset of Movielens1M. The model first adopted three layers of an RBM, and the epoch of each layer of the RBM was set to 10. During the experiment, we found that the error of the third layer RBM increased significantly, which may be due to the overfitting phenomenon. So in the second experiment, we reduced the number of layers of the RBM to two. In order to better indicate the mutation of the third layer when plotting, we set the epoch of each RBM in the second experiment to 15, and the experimental results showed that the DBN formed by the two-layer restricted Boltzmann mechanism worked better. The experimental results are shown in Fig. 5.

As shown in Fig. 5, errors_DBN_2 represents a DBN with two hidden layers, and errors_DBN_3 represents a DBN with three hidden layers. In the 20th iteration, errors_DBN_2 converge gradually, while errors_DBN_3 rise sharply. This is enough to suggest that a DBN with two hidden layers works better.

The choice of learning rate is related to the speed of convergence of our model. The model with two hidden layers is tested on the dataset Movielens 1M, and the learning rate is set as 0.01, 0.075, 0.1, 0.25, 0.5, and 1, respectively. The effect of the model is shown in Fig. 6. In Fig. 6A, different learning rates make the model converge after 15 epochs, and it can be observed that the model with a learning rate of 0.01 is slower. Fig. 6B shows the changes in the model after 15 epochs. It can be seen that the learning ability of the model

Li et al. (2023), *PeerJ Comput. Sci.*, DOI 10.7717/peerj-cs.1448     15/25

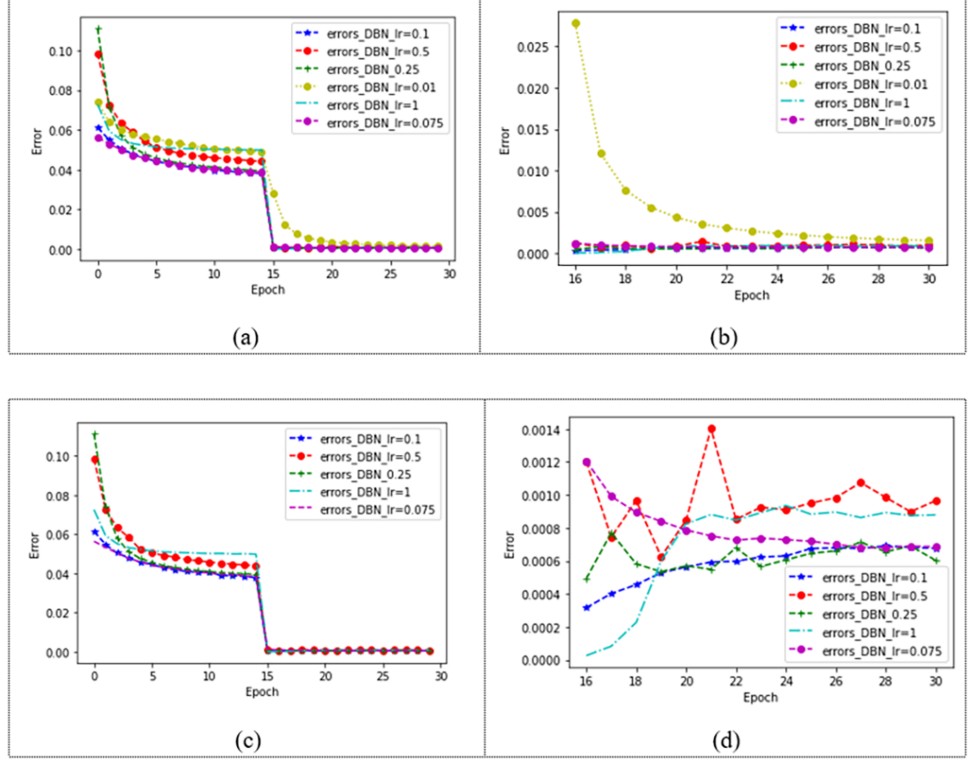

**Figure 6** Convergence of DBN at different learning rates.

**Table 4** Mean absolute error of DBN at different learning rates.

| lr | 0.01 | 0.075 | 0.1 | 0.25 | 0.5 | 1 |
|---|---|---|---|---|---|---|
| MAE | 0.001543 | 0.000679 | 0.000669 | 0.000671 | 0.000826 | 0.000433 |

is poor when the learning rate is 0.01. In order to observe the changes in the model under other learning rates, the learning rate of 0.01 is removed from Fig. 6C. Figure 6C shows the convergence of the model under four different learning rates, and the changes in the model after 15 epochs can be observed in Fig. 6D. When the learning rate is 1, the error of the model is the smallest, and the error gradually increases with the increase of the number of iterations. When convergence, the error of the model with a learning rate of 0.1, 0.25, 0.5, and 1 changes greatly, while when the learning rate is 0.075, the model error decreases with the increase of the number of iterations and finally converges. The error of the model under each learning rate is shown in Table 4.

### Ablation experiment

After the number of hidden layers of the DBN was determined, we started building the predictive model. The whole model was divided into three parts, namely the RBM, the DBN, and the classification model. In this section, three experiments were performed on the Movielens1M dataset to verify the efficiency of the proposed model. The models of the

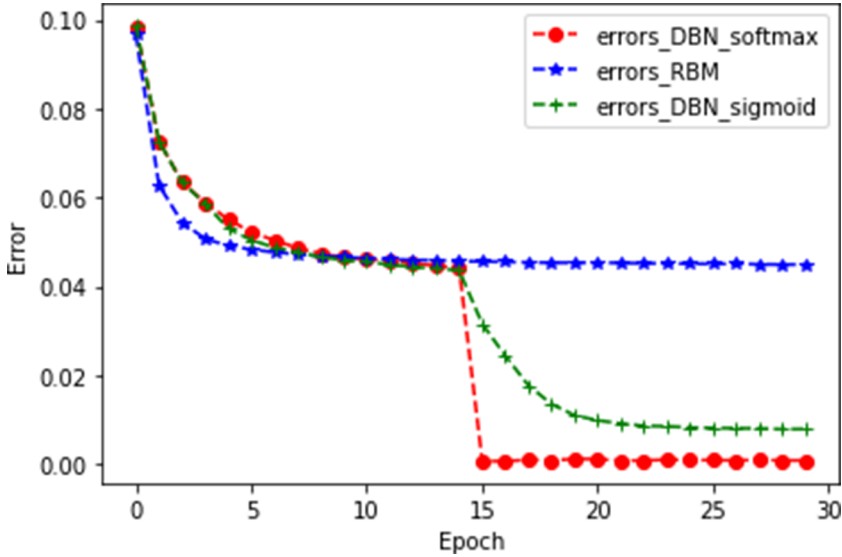

**Figure 7    Ablation experiment.**

**Table 5    The results of ablation experiment and the comparison of the time used by the model, the time unit is seconds.**

| Model | MAE | Time |
|---|---|---|
| DBN+softmax | 0.000679 | 695 |
| RBM | 0.0449 | 405 |
| DBN+sigmoid | 0.007884 | 916 |

three experiments are RBM, DBN+sigmoid, and DBN+softmax. The experimental results are shown in Fig. 7.

As shown in Fig. 7, errors_DBN_softmax represents the DBN+softmax model, errors_RBM represents a single RBM, and errors_DBN_sigmoid represents the DBN+sigmoid model. It can be seen from the figure that the error of DBN+softmax decreases rapidly at 15 epochs, and when it tends to be stable, the MAE value is the smallest. For the training time of the model, the DBN+softmax model is more efficient than the DBN+sigmoid model, while the DBN+softmax model has a longer training time than the RBM model. This is an acceptable time loss because deep neural networks are trained for a longer time than shallow neural networks. However, the DBN+softmax model fits better than the RBM model. The experimental results show that our model is significantly better than the other two compared models. The specific experimental results are shown in Table 5.

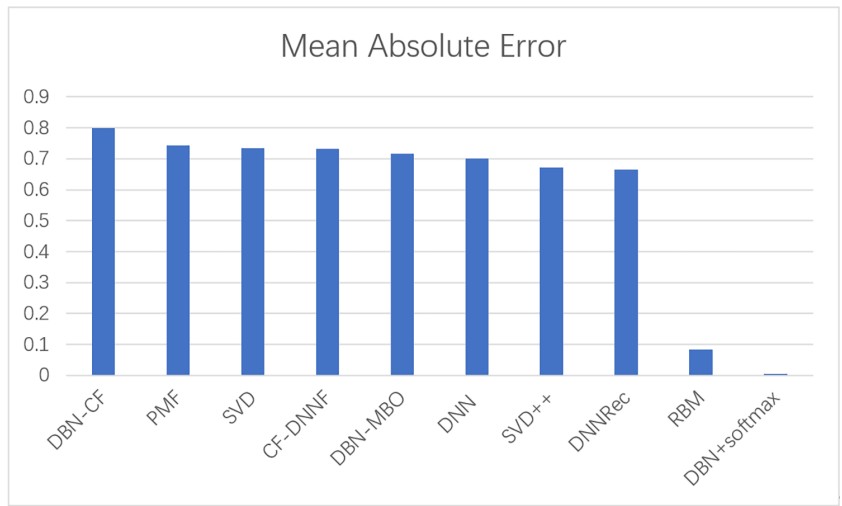

**Figure 8** **A comparison of the MAE values of several models.**

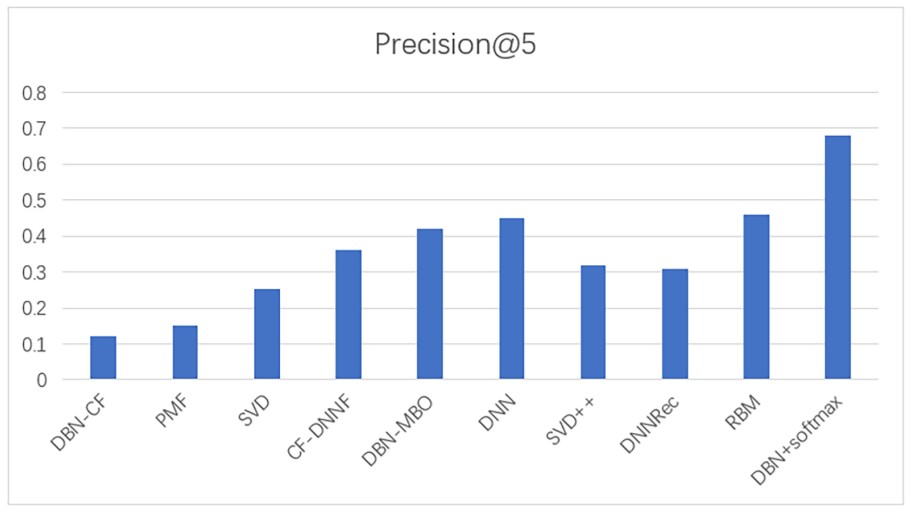

**Figure 9** **A comparison of the precision of several models.**

### Comparison with other models

To prove the effectiveness of our model, the proposed model was compared with other movie recommendation models and experiments were conducted on the Movielens 100K dataset. The results of the experiment are shown in Figs. 8 and 9.

Probabilistic matrix factorization (PMF) (*Mnih & Salakhutdinov, 2007*): A probabilistic matrix factorization model is a collaborative filtering algorithm that can handle very large data sets that performs well on large, sparse datasets.

Singular value decomposition (SVD)++ (*Koren, 2008*): The SVD++ algorithm introduces implicit feedback and user attribute information based on the SVD algorithm, which is equivalent to introducing additional information sources.

Deep neural networks (DNN) (*Liu et al., 2017*): It is established based on the perceptron model, and the training mechanism of layer-by-layer pre-training is adopted on the neural network layer, which solves the shortcomings of traditional neural networks that are easy to overfit and slow training.

SVD (*Zhang et al., 2018*): The algorithm is based on the singular value factorization method, and the use-item rating matrix is broken down into three matrices, which are used for predictions.

DBN-CF (*Ouhbi et al., 2018*): A movie recommendation algorithm based on DBN and item-based collaborative filtering.

CF-DNNF (*Fang, Li & Gao, 2020*): A collaborative filtering movie recommendation algorithm based on deep neural network fusion. The long short-term memory network is used to obtain the characteristic matrix of users and items, and then DBN uses the characteristic matrix to output the probability.

DNNRec (*Kiran, Kumar & Bhasker, 2020*): A movie recommendation algorithm based on hybrid deep learning, which generates ratings for each pair of users and items.

DBN-MBO (*Sridhar, Dhanasekaran & Latha, 2023*): A recommendation system combining a butterfly optimization algorithm and a deep confidence network, this recommendation system mainly recommends movies based on the user's preferences.

In this experiment, MAE values and precision are used as evaluation indicators to evaluate the effectiveness of the model. The above experiments verify the comparison results of our model with commonly used models such as SVD and SVD++. The statistical results are shown in Table 6. The results show that the model based on the traditional collaborative filtering algorithm has a high error, such as SVD and PMF, which may be due to the low utilization rate of the user-item matrix. However, the error of the neural network model or the model based on the combination of neural networks and collaborative filterings, such as DNNRec and RBM, is relatively low, which may be because the neural network has fully learned the features of the user-item scoring matrix. In a word, our model shows great advantages, the MAE value is significantly lower than other algorithms and the accuracy of recommendation is higher than other algorithms. The MAE value is 98% lower than the optimal model RBM. This indicates the usability of our model and its effectiveness in the field of recommendation.

### Dataset size sensitivity analysis experiment

To verify that our model performs well on different-sized datasets, the model was tested on three different sizes of Movielens datasets. And set the batch size to 200 and epochs to 30, where we set the learning rate of the Movielens 100K to 0.125, the learning rate of the Movielens 1M to 0.075, and the learning rate of the Movielens 25M to 1. The experimental results are shown in Fig. 10. The error of the model trained from the three different datasets decreases from 0 to 15 epochs, and the error of the model gradually decreases as the dataset increases. The model tends to converge after 15 epochs, and it can be seen in Fig. 10B that the error of the model tends to be stable after 15 epochs. Moreover, the experimental effect of the model is better on the Movielens 25M datasets, and the MAE value reaches 0.000045.

**Table 6  Comparison with other models.**

| Model | Mean absolute error | Precision |
|---|---|---|
| DBN-CF | 0.8 | 0.12 |
| PMF | 0.743 | 0.15 |
| SVD | 0.735 | 0.254 |
| CF-DNNF | 0.731 | 0.36 |
| DBN-MBO | 0.716 | 0.42 |
| DNN | 0.7 | 0.45 |
| SVD++ | 0.671 | 0.32 |
| DNNRec | 0.666 | 0.31 |
| RBM | 0.084 | 0.46 |
| **DBN+softmax** | **0.00145** | **0.68** |

**Notes.**
The bold values show the data of the model, as proposed in this article.

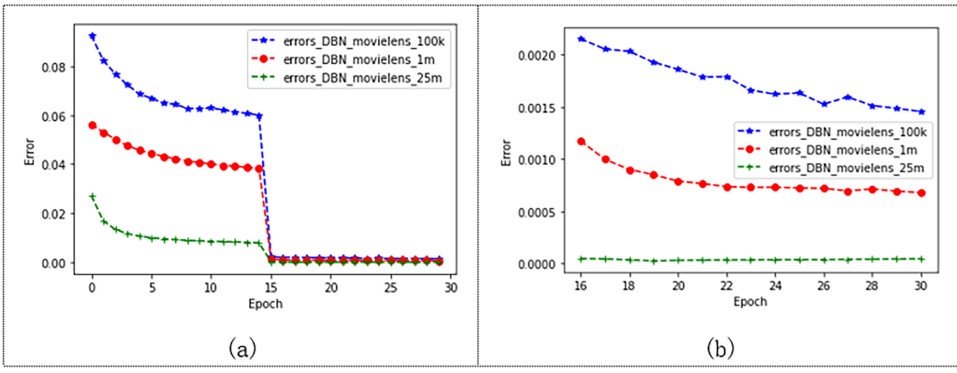

**Figure 10  Performance of the model on three Movielens datasets.**

**Table 7  Experimental results of the model on three Movielens datasets.**

| DataSet | ml-100 lr=0.125 | ml-1m lr=0.075 | ml-25m lr=1 |
|---|---|---|---|
| MAE | 0.001456 | 0.000679 | 0.000045 |

Table 7 shows the results of this experiment. The experimental results show that our model gives better results as the data set increases.

As shown in Fig. 11, we compare the effects of the model on the Douban and Movielens 25M datasets. Douban's dataset has 10,000 users and 1,111 movie ratings. The Movielens 25M dataset has 10,000 users and 24,331 movie records. As shown in the figure, after 15 epochs, the models trained on both datasets tend to converge. The error of the model trained by Movielens 25M converges to 0.000045, and the error of the model trained by the Douban dataset converges to 0.002. Figure 11B shows the convergence state after 15 epochs. The model training error on the Douban dataset is larger, but the model error is

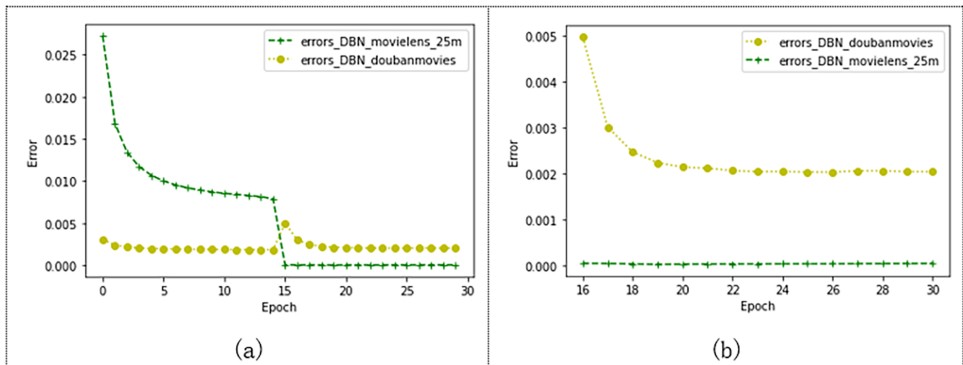

**Figure 11** Effects of the model on Movielens_25m dataset and douban dataset.

also less than 0.003. This may also be related to the larger number of movies cut from the Movielens 25M dataset, which also indicates that our model has more advantages over the model with a larger data volume.

## CONCLUSION

With the large-scale growth of data on the Internet, traditional collaborative filtering techniques that use similarities between users and items to make recommendations have severely compromised the efficiency of recommendations and can be inefficient in the face of sparse data and cold-start problems. However, deep learning can extract the hierarchical feature statistics of users and items to solve the problem of sparse data. Therefore, this article proposes a deep learning algorithm to predict the probability of a user watching a movie. This recommendation algorithm uses a hybrid model based on a DBN and softmax regression to recommend movies. A DBN can deal with massive training data and solve the problem of data sparsity. The extracted user features are input into the softmax multi-classification model, and the negative sampling mechanism is used to further improve the efficiency of the model. Finally, the probability of each user watching each movie is output. Experimental results on the movielens dataset show that the MAE of this model is 98% lower than other models, which is effective for fast and accurate recommendations. In the future, we will train and improve our model on more types of data sets, so that our model can be applied in more fields and spread. It also considers adding an attention mechanism to extract features from sparse data more quickly and efficiently and thus uncover potential interactions between users and items, thereby improving the scalability of the model.

### Funding

This research is funded by the Natural Science Foundation of Hunan Province (nos. 2022JJ50002, 2020JJ6086, and 2021JJ50049) The funders had no role in study design, data collection and analysis, decision to publish, or preparation of the manuscript.

## Grant Disclosures

The following grant information was disclosed by the authors:

Natural Science Foundation of Hunan Province: 2022JJ50002, 2020JJ6086, 2021JJ50049.

## Competing Interests

The authors declare there are no competing interests.

## Author Contributions

- Luyao Li conceived and designed the experiments, performed the experiments, analyzed the data, performed the computation work, prepared figures and/or tables, authored or reviewed drafts of the article, and approved the final draft.
- Hong Huang conceived and designed the experiments, performed the experiments, analyzed the data, performed the computation work, prepared figures and/or tables, authored or reviewed drafts of the article, and approved the final draft.
- Qianqian Li performed the experiments, analyzed the data, authored or reviewed drafts of the article, and approved the final draft.
- Junfeng Man conceived and designed the experiments, performed the experiments, analyzed the data, authored or reviewed drafts of the article, and approved the final draft.

## Data Availability

The data and code are available at GitHub and Zenodo: https://github.com/luyaoli123/recommended-system.

luyaoli123. (2023). Whoyou5069/Code: First release (v1.0.0). Zenodo. Available at https://doi.org/10.5281/zenodo.7807486.

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
