# Peer review of "Personalized movie recommendations based on deep representation learning"

_PeerJ Computer Science, doi:10.7717/peerj-cs.1448_

## Round 0.1 · original submission · Major Revisions

The manuscript presents a model for movie recommendation using a deep belief network (DBN) and a softmax function. The model uses an unlabeled dataset to train the DBN and then fine-tunes the last layer using backpropagation. The probability of a user watching a movie is predicted using the softmax function.

The manuscript provides a clear explanation of the model structure and the reasoning behind its design choices. However, there are a few areas that could be improved:

The manuscript could benefit from more detailed explanation of the bottom-up greedy learning algorithm used to optimize the internal structure of the DBN.

It would be helpful to provide more information about the dataset used to train and test the model, such as the number of users and movies, the sparsity of the data, and the evaluation metrics used to measure the model's performance.

It would be useful to compare the performance of the proposed model to existing models for movie recommendation, in order to demonstrate its effectiveness.

The manuscript could benefit from a more comprehensive discussion of the limitations of the model, such as the computational efficiency and the scalability of the model. Some suggested references: (1) Smartbuddy: defining human behaviors using big data analytics in social internet of things
IEEE Wireless communications 23 (5), 68-74, 2016.
(2) Towards a Personalized Movie Recommendation System: A Deep Learning Approach, ICAIIS 2021.

The manuscript could also benefit from more detailed explanation of the negative sampling softmax used to improve the computational efficiency, and the results of using this technique.

It would be helpful to have a more detailed explanation of the softmax function, including the mathematical details, and how it is used in the model.

Finally, a more detailed description of the implementation of the model, including the programming languages and libraries used, would be beneficial for reproducing the results.

Reviewer 1 ·

Basic reporting

The authors explored a personalized recommendation framework based on deep belief Network (DBN) and softmax regression. The negative sampling mechanism used in this method improved the efficiency of the recommendation. The experimental results show that the performance of this model is better than other common models. Though the presented study is interesting, some necessary revisions are required to improve the manuscript.

1. The authors need to proofread the article once more for English corrections for example the last sentence in “Abstract”.
2. All abbreviations should be defined for the first time not only in the "Abstract" but in the body of the manuscript such as SVD in the last line of “Abstract”.

Experimental design

1. In the second paragraph from the bottom, “In this paper, a DBN based on an RBM was proposed”- Was the DBN model based on an RBM first proposed in this study?
2. Is there any recommendation algorithm based on the DBN model? If so, why not compare the proposed method with other state-of-the-art techniques of DBN?

Validity of the findings

In figure 6 (b) and (d), the numerical value of horizontal coordinate “Epoch” is reasonable?

Additional comments

The references should be updated with some work in 2022.

Reviewer 2 ·

Basic reporting

1. Please rewrite the abstract as follows: 1-2 sentences on the context and the need for the study; several sentences on the model; 2-3 sentences on how the model can be applied and its capabilities; 1-2 sentences on key conclusions and recommendations.
2. The abstract should include quantitative results
3. The current Introduction is too simple, it should include background, current progress, research gaps and the objective of this study, etc (Please emphasize the novelty and impactful contribution of this work as currently this appears to be marginal. The scientific contributions of this study could be further improved).
4. For readers to quickly catch your contribution, it would be better to highlight major difficulties and challenges, and your original achievements to overcome them, in a clearer way in abstract and introduction.
5. Full names should be shown for all abbreviations in their first occurrence in texts.
6. Please cite the corresponding references for the models/equations/formulas that were not originally developed by yourself.

Experimental design

1. Please rewrite the abstract as follows: 1-2 sentences on the context and the need for the study; several sentences on the model; 2-3 sentences on how the model can be applied and its capabilities; 1-2 sentences on key conclusions and recommendations.
2. The abstract should include quantitative results
3. The current Introduction is too simple, it should include background, current progress, research gaps and the objective of this study, etc (Please emphasize the novelty and impactful contribution of this work as currently this appears to be marginal. The scientific contributions of this study could be further improved).
4. For readers to quickly catch your contribution, it would be better to highlight major difficulties and challenges, and your original achievements to overcome them, in a clearer way in abstract and introduction.
5. Full names should be shown for all abbreviations in their first occurrence in texts.
6. Please cite the corresponding references for the models/equations/formulas that were not originally developed by yourself.

Validity of the findings

1. Please rewrite the abstract as follows: 1-2 sentences on the context and the need for the study; several sentences on the model; 2-3 sentences on how the model can be applied and its capabilities; 1-2 sentences on key conclusions and recommendations.
2. The abstract should include quantitative results
3. The current Introduction is too simple, it should include background, current progress, research gaps and the objective of this study, etc (Please emphasize the novelty and impactful contribution of this work as currently this appears to be marginal. The scientific contributions of this study could be further improved).
4. For readers to quickly catch your contribution, it would be better to highlight major difficulties and challenges, and your original achievements to overcome them, in a clearer way in abstract and introduction.
5. Full names should be shown for all abbreviations in their first occurrence in texts.
6. Please cite the corresponding references for the models/equations/formulas that were not originally developed by yourself.

Additional comments

1. Please rewrite the abstract as follows: 1-2 sentences on the context and the need for the study; several sentences on the model; 2-3 sentences on how the model can be applied and its capabilities; 1-2 sentences on key conclusions and recommendations.
2. The abstract should include quantitative results
3. The current Introduction is too simple, it should include background, current progress, research gaps and the objective of this study, etc (Please emphasize the novelty and impactful contribution of this work as currently this appears to be marginal. The scientific contributions of this study could be further improved).
4. For readers to quickly catch your contribution, it would be better to highlight major difficulties and challenges, and your original achievements to overcome them, in a clearer way in abstract and introduction.
5. Full names should be shown for all abbreviations in their first occurrence in texts.
6. Please cite the corresponding references for the models/equations/formulas that were not originally developed by yourself.

---

## Round 0.2 · Minor Revisions

The manuscript presents a model for movie recommendation using a deep belief network (DBN) and a softmax function. The model uses an unlabeled dataset to train the DBN and then fine-tunes the last layer using backpropagation. The probability of a user watching a movie is predicted using the softmax function. The authors has worked significantly and this paper can be accepted, but just some minor issues are to be handled first.

Reviewer 1 ·

Basic reporting

no comment

Experimental design

no comment

Validity of the findings

no comment

Additional comments

no comment

Reviewer 2 ·

Basic reporting

I think the paper is acceptable in its current form.

Experimental design

I think the paper is acceptable in its current form.

Validity of the findings

I think the paper is acceptable in its current form.

Additional comments

I think the paper is acceptable in its current form.

·

Basic reporting

The rebuttal letter provided by the authors includes a detailed explanation of the queries. However, there are a few corrections to be made to the manuscript.

The manuscript has to be proofread again for improving the language structure and avoiding grammatical and punctuation errors.

Line #54: To enhance the professionalism and readability of the text, it is strongly recommended that headings always begin with a capital letter.

In Section 1 and various other sections: the reference formats are mismatched in various places while referring to various research articles. It is recommended to follow the appropriate referencing formats and maintain uniformity throughout the manuscript. Kindly check out this link (https://peerj.com/about/author-instructions/#reference-format) and make corrections.

In Section 4.2.5, the line spacing of the entire sub-section contents is not aligned with the format of previous sub-sections. It would be better to maintain uniformity throughout the manuscript. Kindly update and make corrections accordingly.

In Section 2.4, the mismatching of font should be corrected.

Figures and tables should be provided with detailed descriptions (Title). It is recommended to cross-check again and update wherever necessary.

Experimental design

no comment

Validity of the findings

no comment

·

Basic reporting

No Comments

Experimental design

To evaluate the performance of the proposed algorithm, the mean absolute error is used. The model's performance should be evaluated by measuring the accuracy, precision, recall, F1 score, and AUC-ROC. For a better understanding of the performance of the model, you can also plot the confusion matrix or receiver operating characteristic (ROC) curve.

Validity of the findings

In conclusion, the evaluation of the performance of the proposed algorithm is crucial to determine its effectiveness in making accurate predictions. While the mean absolute error is a useful metric for measuring the overall error in the predictions, it may not provide a comprehensive evaluation of the model's performance.

To gain a better understanding of the model's performance, we recommend that the authors also measure the accuracy, precision, recall, F1 score, and AUC-ROC. These metrics can provide more insights into the model's performance in terms of true positive, false positive, true negative, and false negative predictions. Additionally, plotting the confusion matrix or ROC curve can help visualize the model's performance and identify areas for improvement.

Therefore, we suggest that the authors include these evaluation metrics and visualizations in the conclusion of the paper to provide a more comprehensive evaluation of the proposed algorithm's performance. This will help readers better understand the strengths and weaknesses of the model and its potential for practical applications.

---

## Round 0.3 · accepted · Accept

Thank you for your contribution to PeerJ Computer Science and for addressing the reviewers' suggestions. The reviewers are satisfied with the revised version of your manuscript and it is now ready to be accepted. Congratulations!

·

Basic reporting

Thank you for submitting the revised version of the manuscript entitled "Personalized movie recommendations based on deep representation learning". I have reviewed the changes made in response to the previous comments, and I acknowledge that the authors have addressed the concerns raised. I am satisfied with the changes made and believe that the manuscript is now suitable for publication upon acceptance from the co-reviewers.

Experimental design

no comment

Validity of the findings

no comment